# New insights in atmospheric methane variability in the Arctic by ship-borne measurements during MOSAiC

Amanda Sellmaier[1, 2], Ellen Damm[1], Torsten Sachs[3, 4], Benjamin Kirbus[5, 6], Inge Wiekenkamp[3], Annette Rinke[1], Falk Pätzold[2], Daiki Nomura[7], Astrid Lampert[2], and Markus Rex[1]

[1]Alfred-Wegener-Institute, Helmholtz Center for Polar and Marine Research, Potsdam, Germany
[2]Institute of Flight Guidance, TU Braunschweig, Braunschweig, Germany
[3]GFZ Helmholtz Centre for Geosciences, Telegrafenberg, Potsdam, Germany
[4]Institute of Geoecology, TU Braunschweig, Braunschweig, Germany
[5]Institute for Meteorology, Leipzig University, Leipzig, Germany
[6]Fraunhofer Institute for Energy Economics and Energy System Technology, Kassel, Germany
[7]Hokkaido University, Hakodate, Hokkaido, Japan

Correspondence: Amanda Sellmaier (sam.sellmaier@awi.de)

**Abstract.** The sparse network of Arctic land-based stations results in significant data gaps for atmospheric methane ($CH_4$), particularly in sea-ice covered regions. Ship-based measurements can complement these data, improving understanding of regional and seasonal $CH_4$ variability. This study presents continuous atmospheric ship-borne recordings of $CH_4$ concentration and isotopic composition above the open ocean and sea-ice surface during Leg 4 (June - July 2020) and Leg 5 (August - September 2020) of the Multidisciplinary drifting Observatory for the Study of Arctic Climate (MOSAiC) expedition in the Central Arctic. Our measurements aim to enhance process understanding by identifying local emission sources and refining transport pathway and atmospheric mixing analysis. We compared three contamination-filtering methods and applied the Pollution Detection Algorithm to the raw data. Comparison with nearby land-based stations and their seasonal cycles suggests ship-borne data capture dynamic changes in $CH_4$ sources, sinks, and transport processes, beyond seasonality. To unravel the underlying processes, we identified air mass transport pathways within the atmospheric boundary layer above the Arctic Ocean and their source areas using five-day backward trajectories from the LAGRANTO tool, based on ERA5 wind field data. Our analysis reveals that $CH_4$ variability is driven by air masses predominantly influenced by open ocean and sea-ice-covered regions, with sea-ice dynamics imparting specific modifications along transport pathways. These findings underscore the importance of air mass transport and origin in shaping central Arctic $CH_4$ variability. The study highlights the value of integrating ship-borne $CH_4$ measurements with trajectory analysis to improve process-level understanding and support enhanced regional modelling.

## 1 Introduction

Methane ($CH_4$) is the second most important anthropogenic greenhouse gas after carbon dioxide ($CO_2$) and the atmospheric concentration of $CH_4$ has increased steadily since the pre-industrial era, reaching an global annual mean of 1930 ppb in 2024 (Lan et al., 2025). As the 100-year global warming potential of $CH_4$ is 28 times higher than the one of $CO_2$, the reduction

of $CH_4$ emissions has been identified as a critical factor in limiting climate change (Saunois et al., 2020). A global network of land-based meteorological observation stations operated by the National Oceanic and Atmospheric Administration/ Earth System Research Laboratories (NOAA/ESRL), Advanced Global Atmospheric Gases Experiment (AGAGE), Commonwealth
Scientific and Industrial Research Organisation (CSIRO) and University of California Irvine (UCI) started recording the rise in atmospheric $CH_4$ in the early 1980s (Kirschke et al., 2013). These data can be used to improve top-down estimations and subsequently the modelling of the global $CH_4$ budget (Saunois et al., 2020). By integrating and combining these stations globally, they contribute to a comprehensive understanding of $CH_4$ sources and sinks at both regional and global scales (Berchet et al., 2020; Saunois et al., 2020). These land-based stations have a comparably high level of physical accessibility, however
geographic logistical circumstances often result in considerable distances between stations, particularly at higher latitudes resulting in significant data gaps in marine polar regions (Wittig et al., 2023). Ship-borne recordings of atmospheric $CH_4$ can fill data gaps and hence contribute to improve regional models by providing insights into transport processes and source-sink dynamics in seasonally ice-covered regions (Berchet et al., 2020; Angot et al., 2022a; Pankratova et al., 2022). Here we present ship-borne data recorded during the Multidisciplinary drifting Observatory for the Study of Arctic Climate (MOSAiC) expedi-
tion (Shupe et al., 2022) from June 2020 to October 2020 in the central Arctic Ocean. The three land-based monitoring stations that are located closest to the drift route of the research vessel (RV) Polarstern were Zeppelin Station, Svalbard (ZEP), Alert Station, Canada (ALT) and Tiksi Station, Siberia (TIK) (Fig. 1). We place our dataset in the context of the year-long $CH_4$ concentration measurements reported by Angot et al. (2022a, b).

Atmospheric $CH_4$ in the northern high latitudes arises on the one side from emissions transported from lower latitudes and on
the other side from the existence of a variety of natural and anthropogenic sources, including wetlands, wildfires, permafrost landscapes, and fossil fuel extraction and combustion, localized in the Arctic. Temporal and spatial shifts in the relative contributions of the diverse source emissions induce significant uncertainties in budget estimates (Kirschke et al., 2013; Schuur et al., 2015; Saunois et al., 2017; Turner et al., 2019; Saunois et al., 2020; Wittig et al., 2023). Moreover, efforts to quantify the $CH_4$ budget reveal a notable mismatch between the top-down atmospheric inversions and bottom-up emission estimates
in the Arctic (Kirschke et al., 2013; Saunois et al., 2017, 2020). A method for attributing atmospheric observations to distinct emission sources is analyzing the isotopic signature of $CH_4$ ($\delta^{13}C\text{-}CH_4$). This approach was already applied by Berchet et al. (2020) and Pankratova et al. (2022) for ship-borne measurements conducted in the Arctic Ocean. The isotopic fingerprints of $CH_4$ source emissions can exhibit deviations from the global background signature in the troposphere of around -47.4 ‰ (Pataki et al., 2003; Brownlow et al., 2017; White et al., 2018; Berchet et al., 2020). While emissions from biogenic sources,
such as wetlands, are typically depleted in $^{13}C$ relative to the atmospheric background, with signatures ranging from -70 ‰ to -55 ‰ (France et al., 2016; Nisbet et al., 2016), emissions derived from fossil fuel sources and biomass burning are relatively enriched in $^{13}C$, with signatures between -55 ‰ and -25 ‰ (Kirschke et al., 2013) and -25 ‰ and -13 ‰ (Kirschke et al., 2013; Nisbet et al., 2016), respectively. Our aim is to use continuous ship-based recordings of $\delta^{13}C\text{-}CH_4$ to detect potential local sources and transported $CH_4$ in the marine Arctic. Although sea-ice is assumed to act as a barrier (Parmentier et al., 2013;
Rutgers van der Loeff et al., 2017; Verdugo et al., 2021), $CH_4$ efflux above regions with fractional sea-ice cover, i.e. open leads, has been reported (Kort et al., 2012; Silyakova et al., 2022). Additionally, sea-ice might impact the source and sink balance

during the melt and freeze, respectively (Damm et al. , 2015, 2024). However, our understanding of the physical, chemical and biological processes that might affect $CH_4$ exchange at the ice-covered interface in the Arctic is still very limited.

In addition to local exchange processes, ship-borne recordings capture source emissions transported within air masses, that follow global atmospheric circulation patterns (Kirschke et al., 2013; Saunois et al., 2017). The mobility of ship-based platforms enables the tracking of atmospheric transport and mixing processes of emissions from various sources, providing a key advantage over stationary land-based observatories. As the atmospheric transport pathway lengthens, the initial source signature is altered due to mixing with different air masses, resulting in a mixed isotopic signature (Berchet et al., 2020). Therefore, when analyzing variations in atmospheric time series and attributing them to emission sources, it is essential to consider both the transported source emissions and the effects of transport and mixing processes, as well as potential local emissions (Fleming et al., 2012). A third component affecting atmospheric $CH_4$ levels is the presence of atmospheric sinks, which generate a significant seasonal pattern of $CH_4$ (Schäfer et al., 2016; Lan et al., 2025). The primary atmospheric sink for $CH_4$ is its oxidative reaction with hydroxyl radicals (OH) in the troposphere, which accounts for approximately 90 % of $CH_4$ removal (Kirschke et al., 2013; Rigby et al., 2017; Zhao et al., 2020). The seasonal peak in $CH_4$ concentrations during winter, recorded on all Arctic land-based stations, results from the reduced atmospheric oxidation capacity with lower ultraviolet radiation (Dowd et al., 2023). Remarkably, certain land-based stations exhibit substantial deviations from the expected seasonal pattern (Wittig et al., 2023).

We hypothesize that local emission sources, e.g. from sea-ice-covered regions, and the atmospheric transport of source emissions may contribute to the observed $CH_4$ variations. We suggest that ship-borne recordings in combination with air mass trajectory analysis facilitate the separation of land and marine influences, enabling targeted analysis of marine and cryospheric signals and contributing to the understanding of regional variations within the seasonal patterns.

## 2 Methods

### 2.1 Ship-borne atmospheric $CH_4$ measurements

To measure $CH_4$ concentrations and its isotopic composition in air above the Arctic Ocean during Leg 4 (4 June 2020 – 12 August 2020 with a passive drift in the ice from 19 June, 2020 – 31 July, 2020) and Leg 5 (12 August 2020 – 12 October 2020 with a passive drift in the ice from 21 August 2020 – 20 September 2020) of the MOSAiC expedition, continuous ship-borne measurements were recorded by Cavity Ring-Down Spectroscopy (CRDS) using a Picarro G2132-i isotope analyser (Picarro, Inc., Santa Clara, USA). Air was drawn in from the starboard side of compass deck of RV Polarstern at about 21 m above the sea-ice/water surface using a Teflon tube with a length of 75 m and an inside diameter of 4.5 mm. A flow rate of 2.4 l min$^{-1}$ was generated with a 3KQ Diaphragma pump (Boxer, Ottobeuren, Germany). The Picarro G2132-i isotope analyser then continuously measured in the "High Precision" mode with 1 Hz temporal resolution. The time series presented in this study shows good agreement with the temporal variations in $CH_4$ concentration with data presented in Angot et al. (2022a, b), while the absolute values are consistently about 45 ppb lower. A similar offset magnitude is also evident when comparing our data with the weekly flask measurements from (Dlugokencky et al., 2022). Nevertheless, we decided to publish our dataset

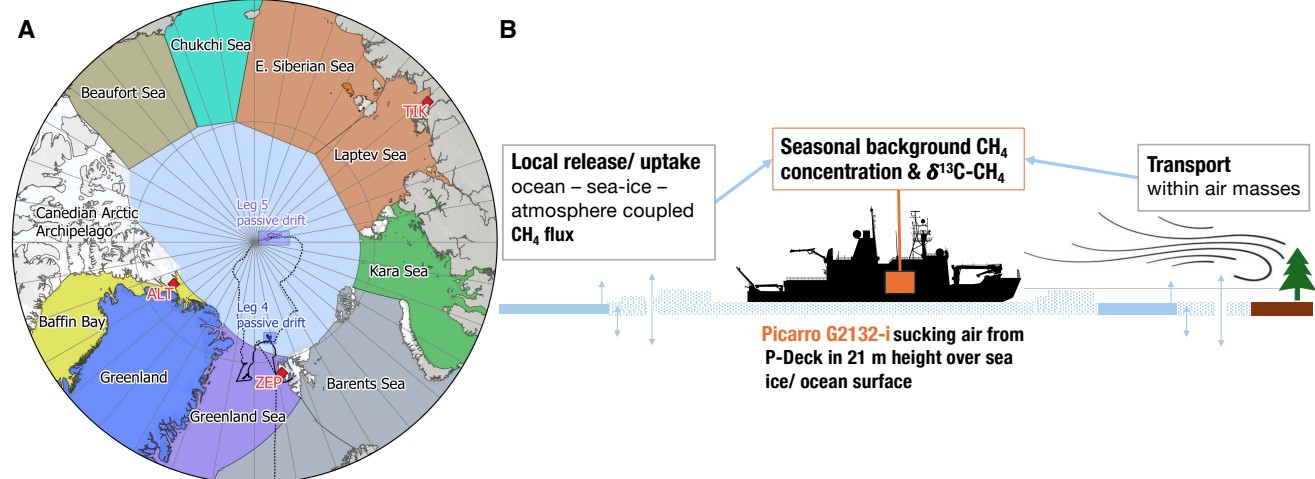

**Figure 1.** (A) Map of the Arctic indicating the ship track of RV Polarstern during MOSAiC (Leg 4, blue; Leg 5, purple). The dashed black lines indicate the whole MOSAiC drift. The color code of the map differentiates nine geographical air mass source areas used in our analysis, the stars represent the closest land-based stations measuring atmospheric $CH_4$: Zeppelin, Ny-Alesund, Svalbard, Norway (ZEP), Alert, Nunavut, Canada (ALT) and Hydrometeorological Observatory of Tiksi, Russia (TIK). (B) Concept of this study, making use of ship-borne $CH_4$ measurements to investigate local release/uptake and long-range transport.

without applying an additional offset correction, as the instrument was calibrated with two reference gases from the Centre for Isotope Research Groningen University, Netherlands (WMO-X2004 calibration scale). Also, our focus is primarily on regional variability rather than absolute concentration levels. In particular, we are interested in relative variations and their relationship to shifts in the isotopic composition. Notably, the presented time series is the only one that simultaneously captures the stable carbon isotopic signature.

## 2.2   Processing of the raw data

In remote locations, observed atmospheric trace gas concentration and isotopic signature measurements may be influenced by local pollution sources from the measurement campaign itself (Beck et al., 2022). A wide range of possible local contamination sources, like pollution from the ship stack, snow groomers, diesel generators, helicopters etc. are described by Beck et al. (2022) and may affect the atmospheric measurements on the ship during the drift with the ice floe. Thus, detecting the effect of local pollution and distinguishing it from the background signal is an important part of processing atmospheric time series (Beck et al., 2022). $CH_4$ is not a significant component of the gases emitted from the stack, but when using the Picarro G2132-i gas analyser, certain cross-sensitivities can occur, appearing as spikes in the recorded $CH_4$ data sets. Three approaches to identify local contamination signals in the time series of $CH_4$ concentration and isotopic composition were compared: spike detecting after Vickers & Mahrt (1997), relative wind direction filter following Beck et al. (2022) and the Pollution Detection Algorithm

(PDA) according to Beck et al. (2022); a comparison of these filter approaches is provided in the Appendix (Fig. A1). Finally, we applied the PDA to the raw $CH_4$ concentration data, as this method most effectively filtered contamination from the data sets without substantial loss of information. The PDA of Beck et al. (2022) was developed based on several atmospheric data sets including the MOSAiC trace gas data set of $CO_2$, with the aim to identify contamination from local pollution sources in aerosol and trace gas data sets recorded in remote regions. The PDA consists of five statistical steps of identifying and flagging

measurement points as local pollution. With this filter approach, 17.3 % of our raw concentration and isotopic signature data was flagged as polluted and consequently removed from the time series.

## 2.3 Five day backwards air mass trajectories

To understand the source regions of the measured air masses onboard RV Polarstern, we used the Lagrangian Analysis Tool (LAGRANTO) to calculate five day backwards air mass trajectories (Sprenger and Wernli, 2015). The essential wind field data are obtained from ERA5 reanalysis with a spatiotemporal resolution of 30 km horizontally and 1 h temporally (Hersbach et al.,

2020; Kirbus et al., 2023). LAGRANTO is a trajectory model that computes air parcel pathways from given wind fields, offering computational efficiency and suitability for identifying air mass origins and transport pathways. Unlike FLEXPART, which additionally accounts for turbulent diffusion, convection, and deposition, LAGRANTO focuses on large-scale flow, thereby reducing model complexity and potential sources of uncertainty. However, both models inherently involve uncertainties. Both

LAGRANTO and the FLEXPART simulations for the MOSAiC campaign (https://webdata.wolke.img.univie.ac.at/mosaic/mosaic.html) are driven by ERA5 reanalysis data, and a comparison between 5-day LAGRANTO and 7-day FLEXPART backward trajectories shows good agreement (see Appendix Fig. B1). The use of LAGRANTO trajectories is well established in Arctic transport studies (Wittig et al., 2023; Kirbus et al., 2023) and provides robust insights into the main transport pathways relevant to this work. Using trajectory analysis, it is possible to track the movement of an air mass backwards in time allowing

the identification of the air mass source area and to characterize the properties of the transport pathway (Fig. 2). The assignment of the geographical air mass source area of the daily backwards trajectories provides the basis for the analysis of the atmospheric time series. At 12 UTC each day, a set of trajectories is initiated at the respective location of RV Polarstern within a 30 km radius and evenly spaced in latitude/longitude every 5 km. The air masses were initiated at 10 hPa above ground level, corresponding to around 80-100 m. The trajectories are then calculated five days backwards in time for the drifting periods during Leg 4 and

Leg 5. For the further analysis we defined nine geographical air mass source areas: the Baffin Bay, Greenland, the Greenland Sea, the Barents Sea, the Kara Sea, the East Siberian Sea/ Laptev Sea, the Chukchi Sea, the Beaufort Sea and the Central Arctic (Fig. 1). The trajectory analysis includes information regarding the geographical source area and sea-ice concentration, number of time steps over land, sea-ice or open ocean along its long-range transport path. As our analysis focuses on interactions with different surface types along the transport pathways, only trajectory portions within the atmospheric boundary layer were

considered. Boundary layer height (BLH) data were extracted from the ERA5 reanalysis, and all trajectory portions above the BLH were excluded from further analysis. To further characterize the air mass transport pathways, sea-ice concentration data were likewise obtained from ERA5 reanalysis. Despite limitations in accurately representing the spatial distribution, including challenges in capturing features like sea-ice leads, the ERA5 data benefits from assimilating diverse satellite observations,

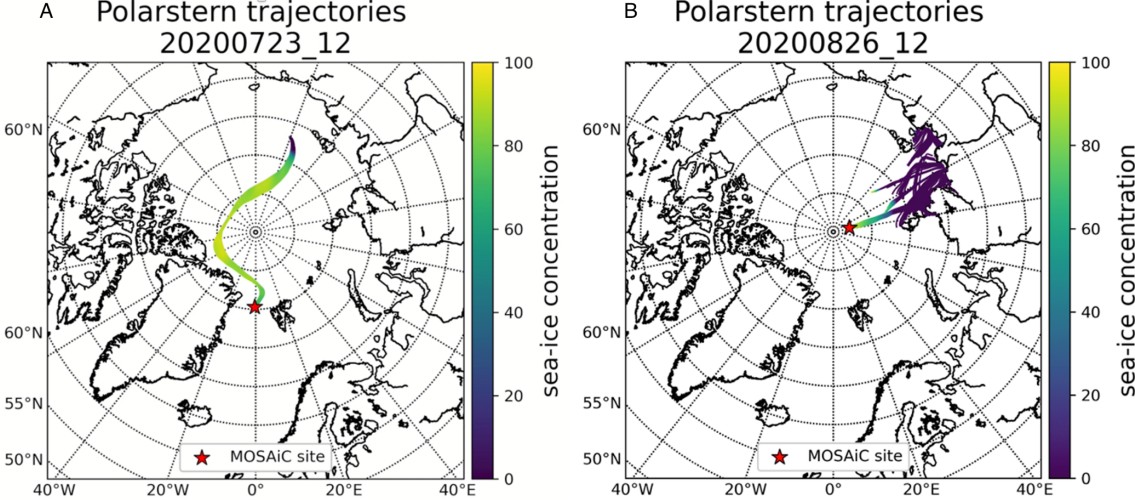

**Figure 2.** Five day backwards air mass trajectories calculated with the LAGRANTO tool based on ERA5 reanalysis wind field data for two exemplary cases, namely (A) 23 July 2020 and (B) 26 August 2020. The colors indicate the sea-ice concentration the air masses are exposed to.

enabling it to approximate the actual sea-ice distribution realistically (Hersbach et al., 2020). Within the atmospheric boundary

layer a constant trace gas mixing ratio is assumed; however, this might not be realistic, as frequent temperature inversions might lead to strong vertical gradients of $CH_4$ concentrations (Lampert et al., 2020).

## 3 Results and discussion

The long-term monitoring data from the nearest Arctic land-based stations along the ship track reveal a distinct seasonal pattern in $CH_4$ concentrations, with elevated levels observed from November to January compared to the period from May to July (Lan

et al., 2025). In January, a significant concentration peak was also recorded near the North Pole recorded during the year-long MOSAiC ice drift expedition (Angot et al., 2022b). This unexpectedly high peak can be attributed to a record-breaking positive phase of the Arctic Oscillation (AO) (Boyer et al., 2023). During the summer season, the most pronounced concentration increase is reported during August (Angot et al., 2022a), aligning with observations from other Arctic land stations (Fig. 3) (Khalil and Rasmussen, 1983; Wittig et al., 2023; Dowd et al., 2023; East et al., 2024) and satellite data (Yurganov et al.,

2021). Furthermore, this study provides an additional ship-borne time series restricted to the summer season (June–July 2020, Leg 4; and August–September 2020, Leg 5) of the MOSAiC ice drift expedition, showing an increase in concentrations in August in the central Arctic Ocean (Fig. 4). In addition to methane concentrations, this study also presents a unique time series of the stable carbon isotopic composition. Remarkably, the TIK station in northeast Siberia exhibits a more heterogeneous and irregular seasonal pattern compared to the stations further west, likely due to the stronger seasonal variations in local $CH_4$

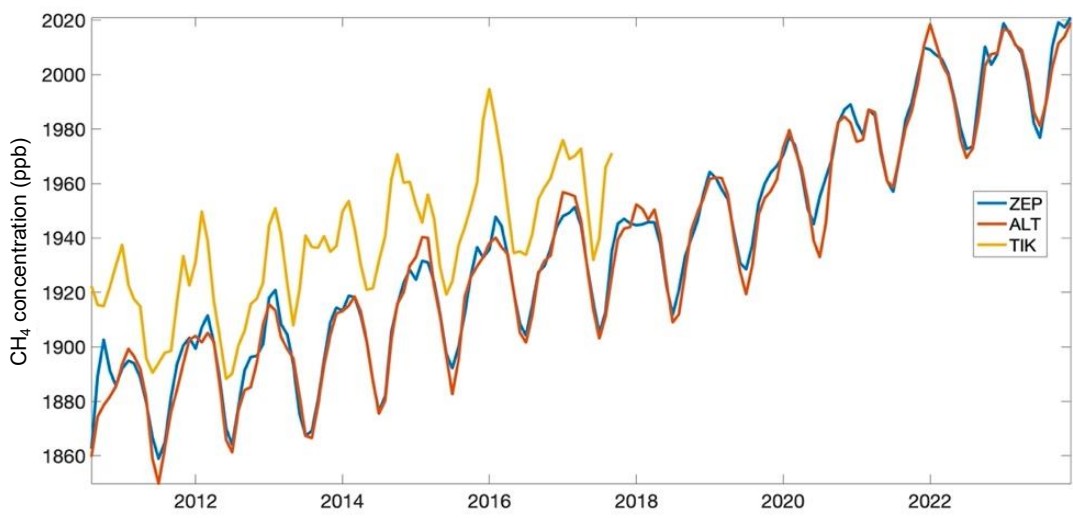

**Figure 3.** Seasonal variations in atmospheric CH$_4$ concentration at three Arctic land stations (Zeppelin, Ny-Alesund, Svalbard, Norway (ZEP), Alert, Nunavut, Canada (ALT) and Hydrometeorological Observatory of Tiksi, Russia (TIK) (period 2011 – 2018). Data from (Lan et al., 2025).

sources, such as thawing permafrost, wetlands, tundra landscapes and wildfires during early summer (Rößger et al., 2022) (Fig. 3). In general, the ship-borne recorded methane concentration and stable carbon isotopic signature range within the expected summer values, while deviations from the seasonal cycle point to additional local to regional processes shaping the pattern of the marine time series (Fig. 4). Hence, we determined first the air mass transport pathways within the atmospheric boundary layer, and second the geographical source area of these air masses.

**3.1  Differentiating marine and terrestrial influences on air masses**

The analysis aims to determine the number of time steps (in seconds) the trajectory travels over land masses, open and sea-ice covered ocean, while the sea-ice concentration corresponds to those during certain time steps. The analysis includes only time steps when the trajectory remains below the atmospheric boundary layer (Fig. 5). In a first step terrestrial and marine influenced air masses were separated from each other (Fig. 5). The terrestrial origin of air masses as well as a terrestrial influence along

their transport pathways were limited to short periods (Fig. 5 and 6A). This absence of terrestrial signals and statistically significant correlations in turn, encouraged a focused analysis of marine signals. Therefore, in a second step, marine air masses were subclassified in those originated over open and sea-ice covered ocean, respectively (Fig. 7). The majority of trajectories (>80 %) evolve over open ocean, while they are subsequently transported over sea-ice before the air masses approach the drifting ice floe at the ship's position (Fig. 5). However, the residence times over sea-ice vary considerably for the individual

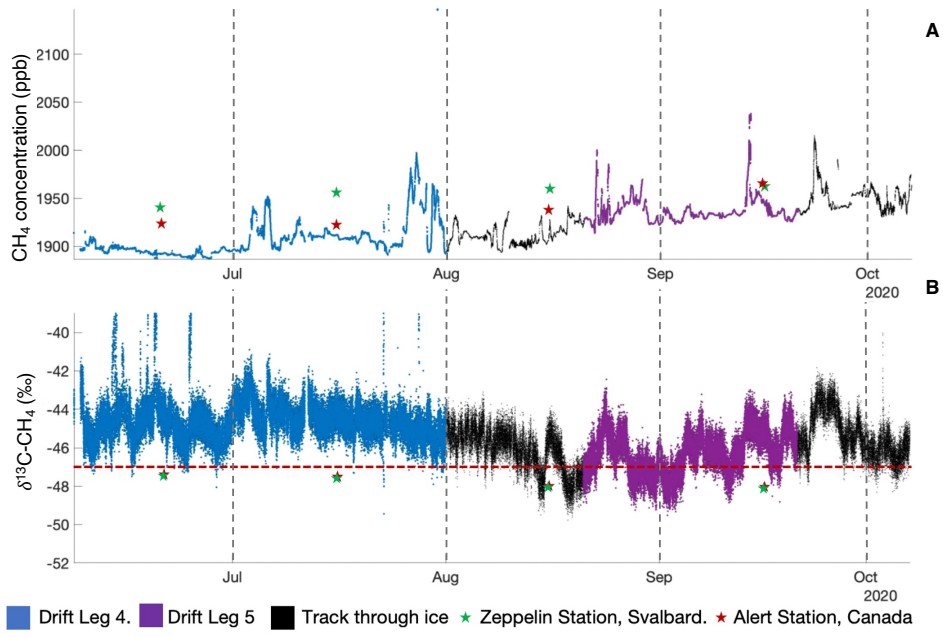

**Figure 4.** Processed ship-borne time series of (A) $CH_4$ concentration and (B) isotopic signature along the track of the RV Polarstern during MOSAiC in the central Arctic. The asterisks indicate the monthly averaged values at Zeppelin Station, Svalbard (green), and at Alert Station, Canada (red) (Lan et al., 2025). The red dotted line represents the atmospheric background isotopic signature of -47.4 ‰ (Lan et al., 2025).

air masses (Figs. 5-7), finally resulting in a mixed, i.e., marine and sea-ice influenced, signal. Distinct surface-specific residence times are partially also induced by the notable spatial distance between the periods of monitoring: The vessel drifted with an ice floe near the sea ice edge close to Spitsbergen during Leg 4, while it was near the North Pole during Leg 5 (Fig. 1)

### 3.2   Methane variability along transport pathways

In the next step we linked surface-specific residence times over marine regions with recorded methane variabilities. According
to (Angot et al., 2022a). the variability of $CH_4$ concentrations remained relatively small in regions close to the North Pole (Leg 5), (Fig. 7B). Remarkable is the limited variability during long-range atmospheric transport over sea-ice with consistently high concentrations (> 50 %) point to a restricted sea-air exchange when the ocean is highly sea-ice covered (Fig. 6 and 7B). By comparison, near the sea-ice edge close to Svalbard (Leg 4) the variability in $CH_4$ concentrations clearly increased locally and with surface-specific residence times (Fig. 7A, B). However, the overall $CH_4$ levels remained lower than those in the
central Arctic, with only individual trajectories showing elevated concentrations (Fig. 7). Near the sea-ice edge, air masses predominantly travelled over sea-ice with generally high sea-ice cover (>50 %) as during Leg 5, and just individual trajectories passed over sea-ice with significantly reduced sea-ice concentrations (10 % - 40 %) (Fig. 7C). Therefore, on the one hand the variability in air masses consistently passing high sea-ice concentrations is low. On the other hand, the methane variability

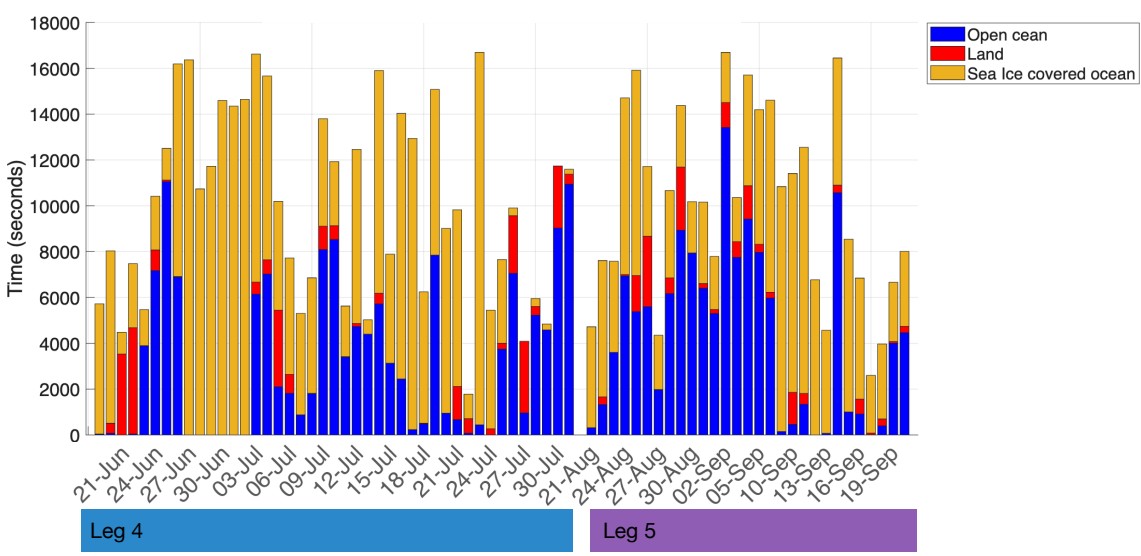

**Figure 5.** Daily air mass backwards trajectories of the drifting periods during Leg 4 and Leg 5 colored in the residence times over land, open and sea-ice covered ocean, respectively. Bar heights indicate the time each trajectory spent within the atmospheric boundary layer.

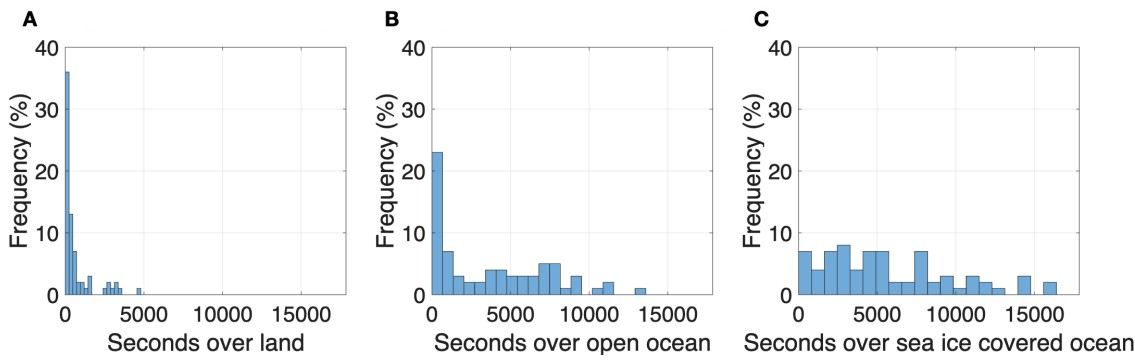

**Figure 6.** The total number of time steps time over land (A), open (B) and sea-ice covered ocean (C), respectively, correlated to the frequency of the daily air mass backwards trajectory during Leg 4 and Leg 5.

in air masses encountering reduced sea-ice cover may be influenced by ice melt near the ice edge. Differences in variability between both legs are also shown by (Angot et al., 2022a). Therefore, we postulate that enhanced local methane levels at the ice edge around Svalbard (Leg 4) related to the region close to the north pole (Fig. 4) may be caused by the seawater which is permanently oversaturated with respect to the atmospheric background (Damm et al. , 2007; Mau et al., 2017; Silyakova et al., 2022). In addition, methane formerly trapped in sea-ice may also be directly emitted during the early melt season (Kort et al.,

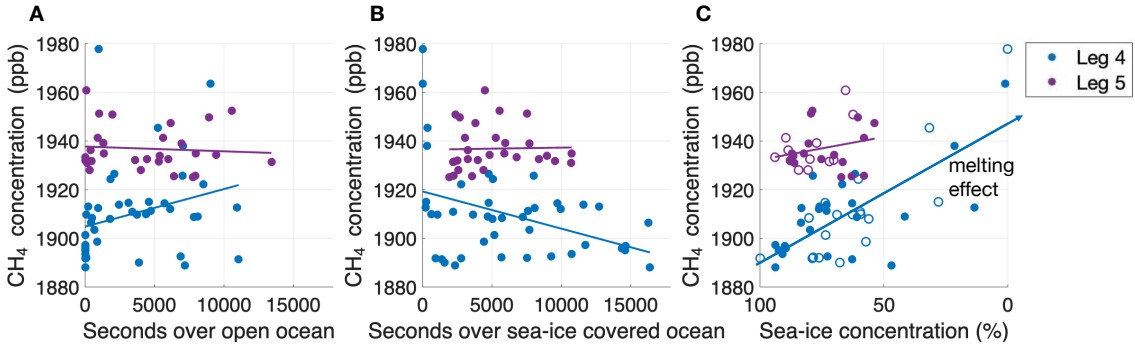

**Figure 7.** Residence time over open (A) and sea-ice covered ocean (B), respectively, correlated to the methane concentration during both legs. C: sea-ice concentration on the air mass transport pathway related to the $CH_4$ concentration. Filled markers indicate trajectories that remained below the boundary layer height (BLH) for $\geq$50 % of their total duration, while unfilled markers represent <50 %.

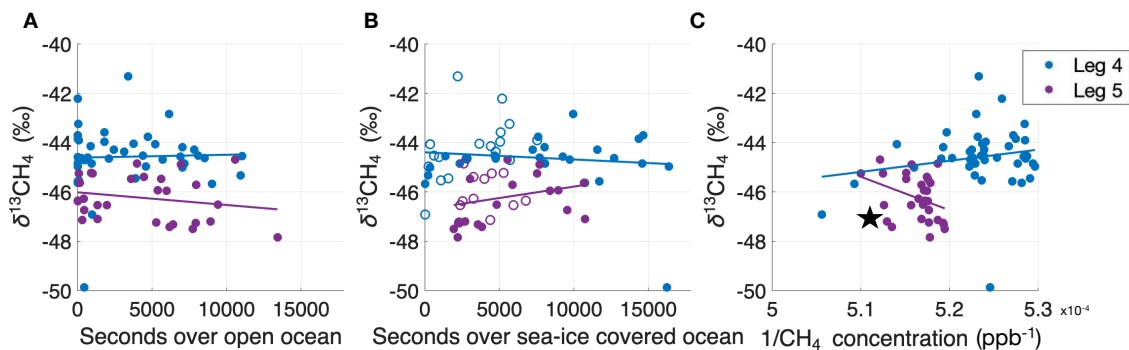

**Figure 8.** Residence time over open (A) and sea-ice covered ocean (B), respectively, correlated to the $\delta^{13}$C-$CH_4$ data during both legs. Filled markers indicate trajectories that remained below the boundary layer height (BLH) for $\geq$50 % of their total duration, while unfilled markers represent <50 %. C: The inverse of the methane concentration related to the $\delta^{13}$C-$CH_4$ data for both legs. The star indicates the atmospheric background, defined by the average $CH_4$ concentration from nearby land-based stations during the measurement period and an isotopic composition of -47.4 ‰ (Lan et al., 2025).

2012; Verdugo et al., 2021). Hence, the increased variability reflects the impact of both, the ice melt and regional differences along certain trajectories. We further examined whether the fraction of time that the air mass trajectories spent below the boundary layer height influenced the observed concentration variability (Fig. 7C). However, given the limited dataset, the results suggest that regional effects exert a stronger influence. A higher data resolution, including all-seasons observations, is needed to reveal how processes coupled to the sea-ice cycle affect variations of the atmospheric methane levels in the Arctic.

### 3.3 Stable carbon isotopic signatures

Also, the stable carbon isotopic signature of $CH_4$ in the Arctic atmosphere reveals a clear seasonal trend, marked by a gradual depletion in $^{13}C$ relative to the atmospheric background value of -47.4 ‰ (Pataki et al., 2003; Brownlow et al., 2017; White et al., 2018; Berchet et al., 2020) throughout the summer months (Platt et al., 2022). The recorded ship's time series reflected this seasonal pattern as well (Fig 3). However, in addition to this overall trend, region-specific divisions described above exposed further pattern (Fig 8). Close to the North Pole (Leg 5), the isotopic signature varied less and corresponded to background values. This concentration pattern corroborates the predominance of transport, unaffected by an exchange with both the open and sea-ice covered ocean (Fig 7A, B). By comparison, near the sea-ice edge (Leg 4), the isotopic signature tends to be slightly enriched in $^{13}C$ relative to the atmospheric background (Fig. 3), while also a significant variability is shown (Fig. 8C). The pronounced shift even exceeded up to 5‰ of the $^{13}C$-enrichment, that is reported during the melt season (Damm et al. , 2015; Verdugo et al., 2021). Further, tracing the impact of the life cycle of sea-ice related to the marine methane pool in a process study revealed shifts in the isoptopic signature induced by kinetic fractionation (Damm et al. , 2024). However, larger data sets are needed to upscale these observations of ice induced fractionation effects to isotopic shifts in the atmospheric isotopic signature.

### 3.4 Variability of $CH_4$ across geographical air masses source areas

To further investigate the initial air mass signal, we assigned the marine time series to predefined source regions based on the trajectory's geographical origin (Fig. 1 and 9). Especially during the first week of Leg 4, most of the air masses originated from the Greenland Sea, while during the rest of the leg more short-term variations in the source areas of the air masses were detected. Air masses from the East Siberian Sea/Laptev Sea had the highest occurrence (frequency of occurrence 34.5 %) during Leg 5. Source area specific differences were revealed, with pronounced concentration peaks primarily associated with air masses originating from the Kara and Barents Seas (Fig. 9 and 10), suggesting these regions as potential marine $CH_4$ source areas, consistent with previous findings (Pankratova et al., 2022).

$CH_4$ concentrations and their isotopic signatures exhibited variations between source areas and the two legs (Fig. 10), with elevated standard deviations for some regions of origin indicating a complex interplay of consistent and variable $CH_4$ sources (Berchet et al., 2020). However, the measurements were conducted at a considerable distance from the respective source regions (Fig. 3) and the initial source signals were altered along their long-range transport path by both dilution and regional $CH_4$ emissions. Therefore, the observed differences between Leg 4 and Leg 5 across all source areas likely reflect a combination of their distinct spatial locations and associated variability in transport-related alterations and potentially seasonal changes. However, the limited knowledge of regional $CH_4$ sources and their seasonality introduces further uncertainty in quantifying specific source contributions (Dowd et al., 2023). Consequently, the applicability of source attribution methods like Keeling plots is limited for our time series.

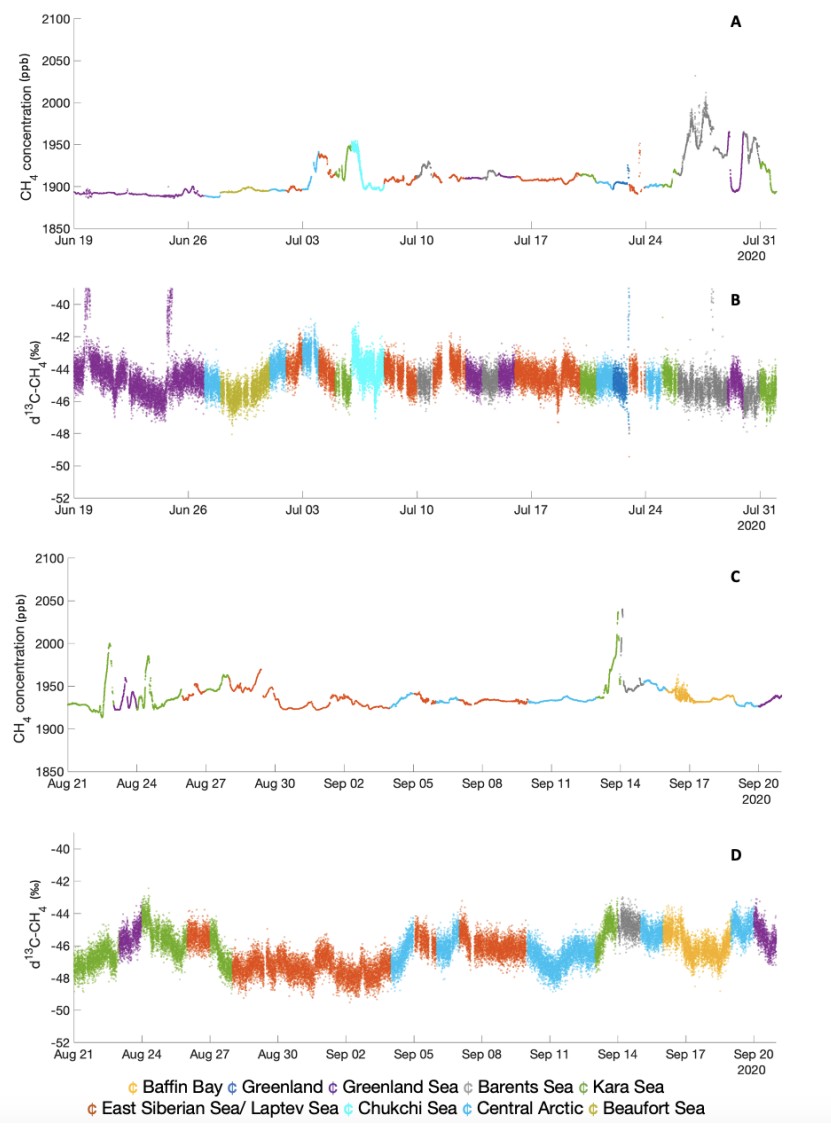

CH$_4$

**Figure 9.** Time series of CH$_4$ concentration and $\delta^{13}$C-CH$_4$ for Leg 4 (A and B, respectively) and Leg 5 (C and D, respectively) coloured in the assigned air mass source area (based on colour code map Fig.1).

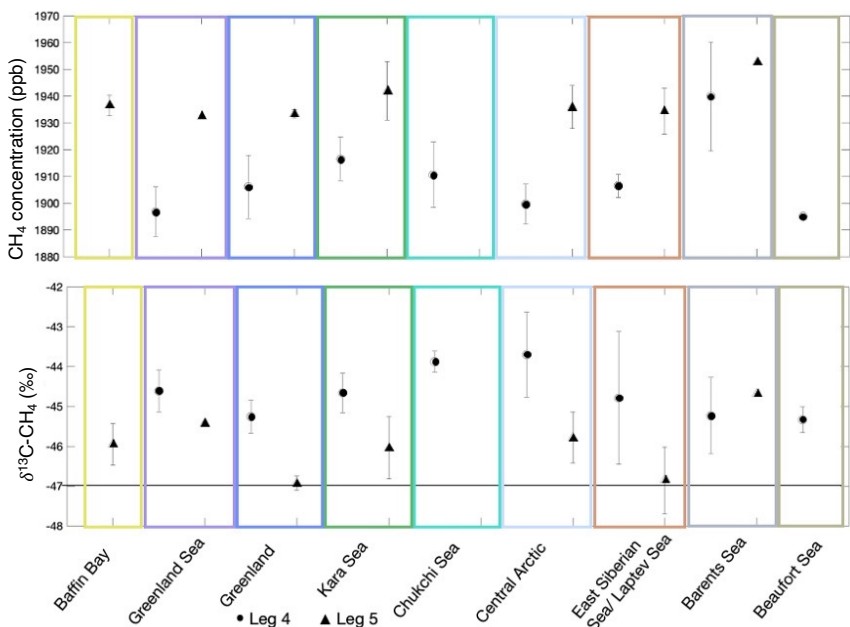

**Figure 10.** Variations between air mass source areas and drifting periods of Leg 4 and 5 in the median and standard deviation of the (A) CH$_4$ concentration and (B) $\delta^{13}$C-CH$_4$. The $\delta^{13}$C-CH$_4$ data are shown in comparison to the atmospheric background level of -47.4 ‰.

## 4  Summary and conclusions

The study provides new insights into the spatiotemporal variability of atmospheric CH$_4$ concentration and its stable carbon isotopic composition in the central Arctic Ocean. While the CH$_4$ concentrations follow the typical seasonal pattern in the high northern latitudes observed in previous studies, the isotopic composition shows deviations that indicate regionally restricted and still unconsidered shifts.

The marine time series combined with air mass backwards trajectory analysis and its residence time within the atmospheric boundary layer enabled an effective distinction between terrestrial and marine air masses, as well as between open ocean and sea-ice covered conditions along the transport path. Low variability in methane concentration near the North Pole (Leg 5) indicates limited sea-air exchange under persistent high sea-ice coverage whereas enhanced variability near the sea-ice edge close to Spitsbergen (Leg 4) refer to impacts by sea-ice melt in a region with known methane excess in seawater. A more robust statistical assessment, however, requires larger datasets that combine trajectories remaining predominantly within the boundary layer with signals modified by air-sea exchange processes. Air mass attribution to geographical source regions indicates elevated CH$_4$ concentrations above the Barents and Kara Sea, yet uncertainties in regional source strength, seasonality and long-range transport constrain robust source attribution.

These findings highlight the need for higher spatiotemporal resolution, including year-round observations, to unravel the complex interplay of sea-ice dynamics and methane cycling. The integration of ship-based measurements with trajectory and

boundary layer analysis provides a valuable framework for addressing observational gaps in remote, sea-ice-covered regions by detecting potential local sea-ice related $CH_4$ exchange processes and supporting regional modelling through insights into atmospheric transport pathways and mixing processes.

## Appendix A:  Comparison of three different pollution filtering approaches

## Appendix B:  Comparison of LAGRANTO and FLEXPART air mass backwards trajectories

*Data availability.*  The final link to the PANGAEA dataset, containing all relevant data, will be provided upon publication.

*Author contributions.*  AS analyzed the atmospheric methane data in the framework of her master thesis supervised by ED and MR, wrote the initial draft and provided the figures. ED, FP, DN and TS participated in the MOSAiC experiment and collected the data. AR and BK supported with the backward trajectory analysis. TS, IW, FP and AL supported the data processing. All authors contributed to revising the text.

*Competing interests.*  The authors declare no competing interests.

*Acknowledgements.*  This manuscript is part of the international Multidisciplinary drifting Observatory for the Study of the Arctic Climate (MOSAiC) with the tag MOSAiC20192020 and the Project ID AWI_PS122_00. We thank the cruise participants, ship's crew and logistics support as well as everyone else who contributed to the realization of MOSAiC (Nixdorf et al., 2021). IW was funded by DFG project number 414169436 and 465048505. Trajectory analysis from the FLEXPART model simulations, performed by the FLEXPART group at the University of Vienna (https://img.univie.ac.at/webdata/mosaic). Special thanks to Silvia Bucci for providing the FLEXPART plots for the comparison.

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

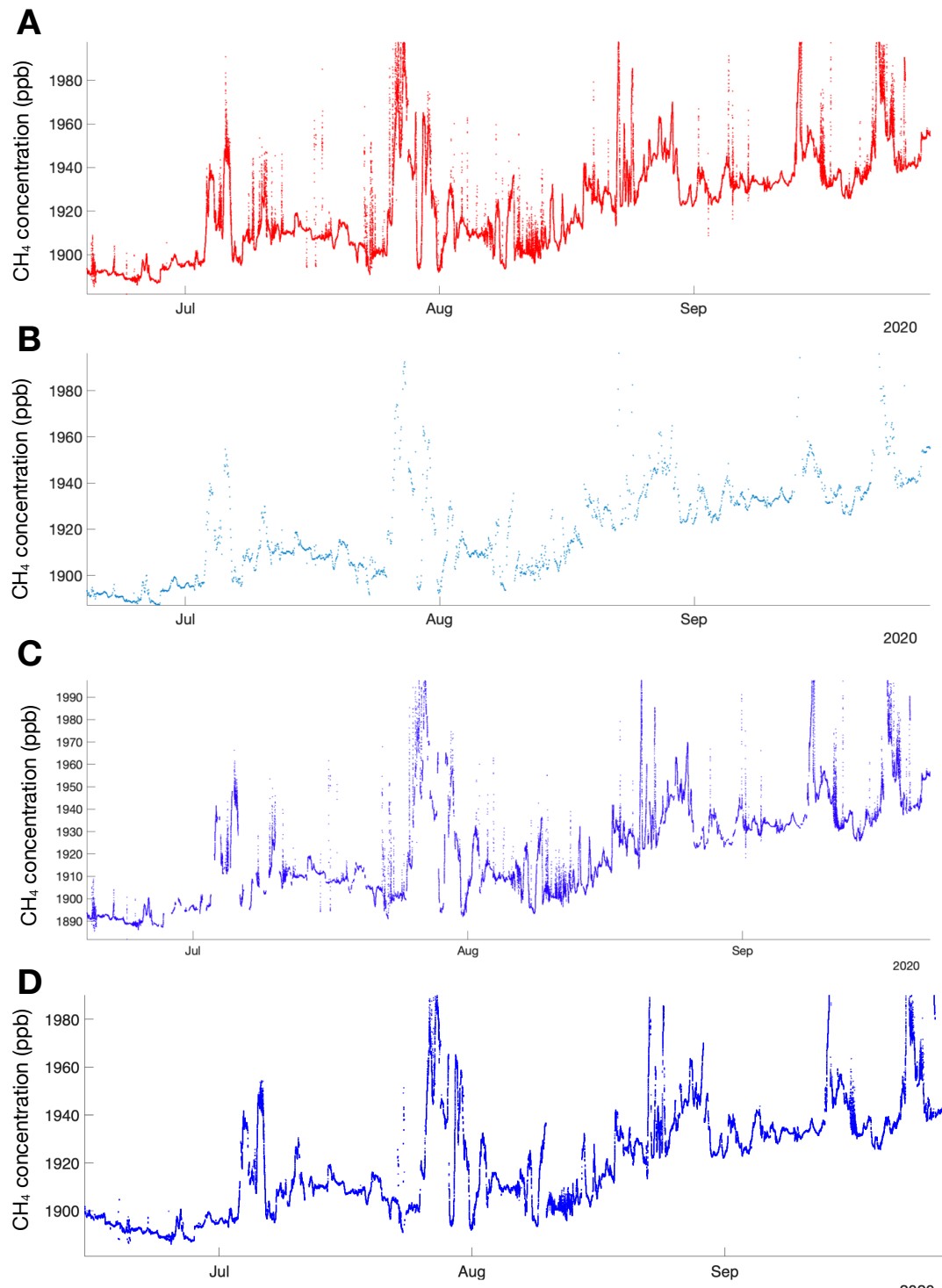

**Figure A1.** Example of the processed CH$_4$ concentration data compared to the raw data showing the resulting smooting and time dimension problem when applying the filters. A: Raw data, B: Spike detection according to Vickers & Mahrt (1997), C: Wind direction filter based on Beck et al. (2022).

, D: Pollution Detection Algorithm after Beck et al. (2022)

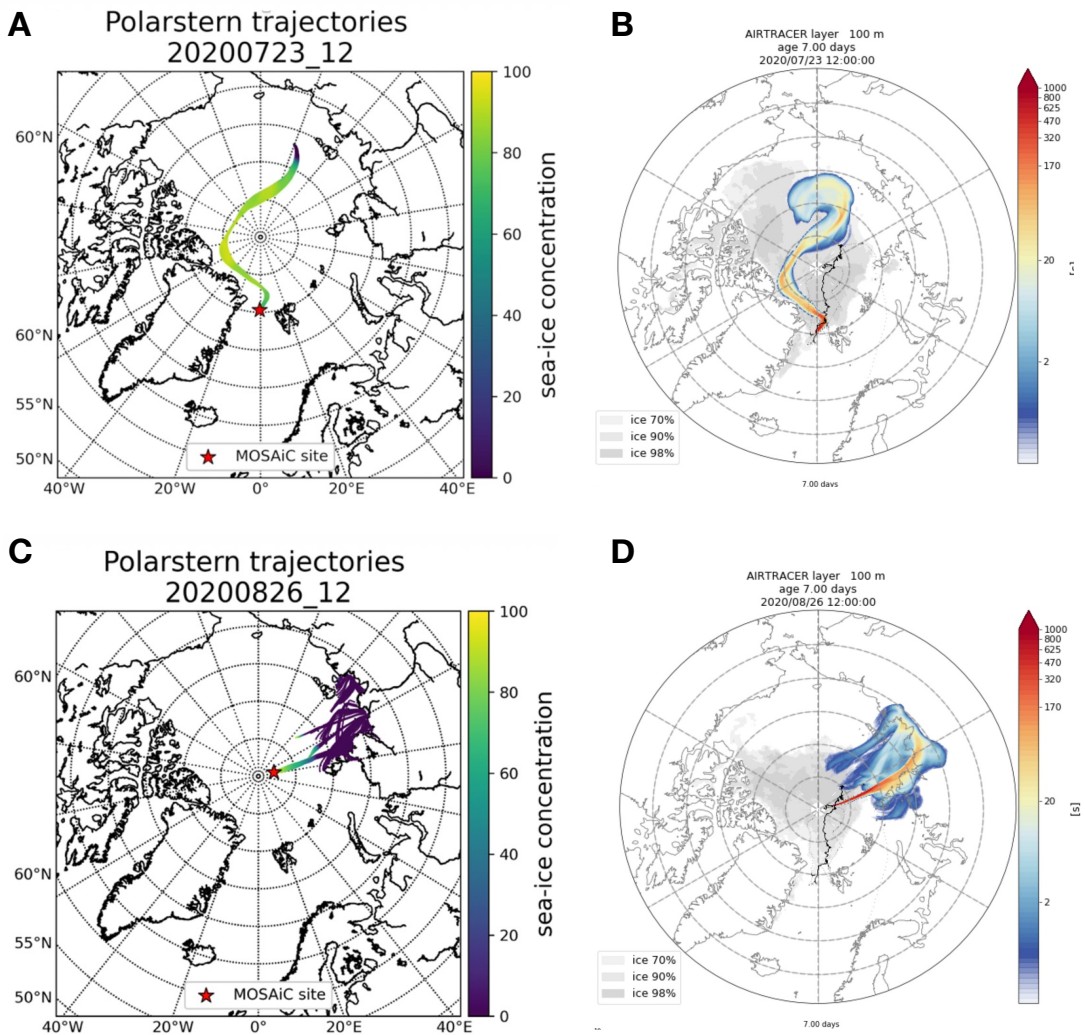

**Figure B1.** Comparison of 5-day air-mass backward trajectories from LAGRANTO (A and C) and 7-day trajectories from FLEXPART (B and D) for two example days during the MOSAiC campaign. Both models are driven by ERA5 reanalysis data. FLEXPART, a full particle dispersion model, includes turbulent and diffusive transport processes, whereas LAGRANTO computes trajectories based solely on resolved wind fields. Despite these methodological differences and trajectory lengths, both models show strong agreement in the large-scale air-mass pathways. Plots of FLEXPART trajectory simulations provided by the FLEXPART group, University of Vienna (https://img.univie.ac.at/webdata/mosaic).