# Peer review of "New insights in atmospheric methane variability in the Arctic by ship-borne measurements during MOSAiC"

_EGUsphere, 2025_

## Author Response (AR1)

**Reviews**

**#1**

This manuscript presents continuous ship-borne measurements of atmospheric methane ($CH_4$) concentration and isotopic composition during MOSAiC legs 4 and 5, complemented by daily backward trajectories to investigate transport pathways and source regions. The study is interesting and contributes valuable observations from an under-sampled region, where land-based stations are sparse and ship-based records can indeed help resolve spatial and temporal variability. The integration of $CH_4$ isotopes with transport analysis has potential to advance our understanding of central Arctic $CH_4$ dynamics.

However, I see a major limitation: the study does not sufficiently connect with, or make use of, the broader suite of datasets that were collected during MOSAiC. This weakens the contextualization and originality of the work. In particular:

1. Existing $CH_4$ datasets from MOSAiC

The authors repeatedly highlight the "need for increased spatiotemporal data resolution, including year-round measurements". These datasets already exist. In Angot et al. (2022), continuous year-round atmospheric $CH_4$ measurements from MOSAiC are reported and made publicly available via PANGAEA (Angot *et al.*, 2022). Acknowledging, referencing, and ideally comparing with these existing records would considerably strengthen the manuscript and properly situate this study within the broader MOSAiC framework.

**Response**

**Thank you for pointing this out. We improved the Introduction by the integration of further year-long datasets (lines 37- 38). Further we included Angot et al. (2022) in the References (line 259 -260). Indeed, we missed this reference by transferring the text to Overleaf. We included the comparison of the datasets in the Methods (lines 86 – 89). We also improved the Results and Discussion section by more effectively integrating our dataset with the year-long observations reported by Angot et al. (2022) and other related measurements (lines 143–150, 175, and 185).**

2. Air mass transport and trajectory analysis

The manuscript relies on daily back trajectories from LAGRANTO. While this provides some insight, a much richer dataset is available: high-resolution FLEXPART footprints and multiple trajectories per day, including emission sensitivities over land, ocean, and sea ice (see https://webdata.wolke.img.univie.ac.at/mosaic/mosaic.html). Incorporating

or at least discussing this dataset would provide a much more detailed and robust analysis of air mass origins and transport, compared to the current daily trajectory approach.

**Response**

**Thank you very much for this important comment. ERA5 reanalysis data are commonly used to drive both the air-mass backward trajectories modeled with LAGRANTO in our study and those available from FLEXPART (https://webdata.wolke.img.univie.ac.at/mosaic/mosaic.html). We acknowledge that the FLEXPART simulations for MOSAiC provide a richer dataset, including high-resolution footprints, multiple trajectories per day, and emission sensitivities over land, ocean, and sea ice. However, FLEXPART is a full particle dispersion model that accounts for additional processes such as turbulence, diffusion, and deposition, while LAGRANTO is a trajectory model that focuses on computing flow paths under given wind fields. In our study, we used LAGRANTO to calculate 5-day backward trajectories, whereas the FLEXPART dataset provides 7-day trajectories. The relative simplicity of LAGRANTO makes it computationally efficient and particularly well suited for source attribution, atmospheric transport studies, and identifying air mass origins. While it does not represent diffusive and turbulent dispersion in detail, its reduced structural complexity also implies fewer sources of model uncertainty. To further support this, we attach a comparison of the two model outcomes for two example days, both of which show good accordance between LAGRANTO and FLEXPART (Appendix Fig. B1, line 262). Thus, although we rely on LAGRANTO trajectories, their robustness has been demonstrated in the literature, including in Arctic applications (Wendisch et al., 2024; Kirbus et al., 2024), giving confidence that our analysis captures the main transport pathways relevant for this study. We updated the Methods chapter (lines 116-124) and included a comparison of the LAGRANTO and FLEXPART air mass backwards trajectories (in the Appendix as Fig. B1).**

Additional comments

3. Calibration of the instrument

The Methods section does not describe the calibration procedures used during the expedition. Furthermore, the authors state that "mean $CH_4$ concentrations were approximately 5% lower than the simultaneous measured mean levels at the nearby land-based stations". This is interpreted as a potential dilution of the signal during long-range transport over sea ice. However, the reported concentrations appear low compared to the merged MOSAiC dataset (Angot *et al.*, 2022), which is traceable to the WMO-X2004A calibration scale. A direct intercomparison would help clarify whether this difference reflects calibration issues, transport effects, or other causes.

**Response**

**We appreciate the reviewer's comment and are aware of this point. Our focus during MOSAiC was on local ice–water–atmosphere exchange processes. The presented dataset is intended as a process study to improve understanding of surface interactions. Therefore, this work complements, rather than duplicates, the study by Angot et al. 2022.**

**During the winter months, our Picarro was exclusively used for measurements of methane concentration and isotopic composition in water and sea ice. While these measurements did not occur during the summer month due to missing man power, we decided to run the Picarro onboard for atmospheric measurements instead of shutting it down.**

**We used the Picarro G2132-i isotope analyser, the instrument was calibrated with two reference gases from the Centre for Isotope Research Groningen University, Netherlands (WMO-X2004 calibration scale) (added to Methods line 89 – 93). We agree that an intercalibration of different Picarro instrument types used by Angot et al. (2022) and the Picarro G2132-i isotope analyser will be beneficial for future research.**

**We revised the Results and Discussion section to focus consistently on variations rather than comparisons of absolute values (lines 174–179 and 182–**

**190), and applied the same approach in the Summary and Conclusion section (lines 232–234).**

4. Pollution detection algorithm

The authors note that three contamination-filtering approaches were tested, and that the Beck et al. (2022) Pollution Detection Algorithm was ultimately chosen. Since pollution filtering is a critical and often challenging step, I would encourage the authors to show these results, at least in the Supplementary Information. For example, comparing datasets flagged by different methods would provide a useful sanity check. It would also be valuable to compare the final QC'd dataset with that of Angot et al. (2022) to assess consistency between independently collected and cleaned datasets.

**Response**

**We have included a new figure in the Appendix (Fig. A1, line 257) comparing the results of the three contamination-filtering approaches compared to the raw methane concentration data. This comparison illustrates the differences between the methods and supports our choice of the Pollution Detection Algorithm (Beck et al., 2022) as the most suitable approach for our analysis.**

5. Seasonal cycle and short-term variability

The manuscript reports a "most pronounced concentration increase during August", relatively stable concentrations near the North Pole (leg 5), and higher variability close to Svalbard (leg 4). How does this compare with the year-round dataset reported by Angot et al. (2022) (see attached Figure showing monthly distributions)? For example, that dataset shows a large spike in early January (contextualized in Boyer et al. (2023)), and higher variability in July–September compared to the autumn and spring periods. A discussion linking the ship-borne observations to this broader seasonal context would be highly informative.

**Response**

**We appreciate the reviewer's comment and agree that linking our observations to the broader seasonal context of Angot et al. (2022) provides important insight. We have therefore expanded the discussion accordingly (Lines 143–150).**

**6. Isotopic composition**

The isotopic data are of great interest but appear somewhat underutilized. I would have expected a more detailed discussion of possible source contributions, such as Siberian anthropogenic point sources or wildfire plumes (e.g., the July 27–29 event). Further interpretation of the isotopic signatures could add significant value to the manuscript.

**Response**

**Our trajectory analysis shows that terrestrial air mass origins and corresponding influences along the transport pathways were restricted to relatively short periods. Thus, the influence of land-based sources, such as Siberian anthropogenic emissions or wildfire plumes, is not detectable in the central Arctic during the period covered by our observations. Given the limited dataset, our intention was to provide an initial assessment of how isotopic signatures can support the interpretation of air-sea/ice exchange in the central Arctic and and help to distinguish these from background variability, without overinterpreting the available data. Further, combining trajectory analysis with boundary layer height information will improve to distinguish local and regional signals from large-scale background variabilities in future studies.**

Recommendation: This study provides valuable data, but in its current form it underutilizes the wealth of complementary MOSAiC datasets. A revision that (i) acknowledges and integrates year-round $CH_4$ observations, (ii) considers high-resolution transport datasets, and (iii) provides more detailed discussion of calibration, pollution filtering, and isotopic signatures would substantially improve the robustness, originality, and impact of the work.

**Response**

**We thank the reviewer for their thorough revision and many helpful suggestions to improve our study. We integrated year-round $CH_4$ observations and high-resolution transport datasets (see: lines 37, 38 and lines 86 – 89).**
**We included a comparison of the two air mass backwards trajectory modelling tools LAGRANTO and FLEXPART (lines 114-121 and Appendix Fig. B1) and**

**showed that our analysis with LAGRANTO captures the main transport pathways relevant for this study.**

**We provided a more detailed discussion on pollution filtering (Appendix Fig. A1, Methods line 105) and calibration (Methods lines 89 – 93).**

**We explained our point of view regarding the discussion of the isotopic signature data (see response above).**

**#2**

This is a well-written paper on a potentially informative and important topic, especially so given the limited availability of methane time-series data from the Arctic Ocean.  This manuscript relies substantially on the interpretation of airmass back trajectories and my only significant concern deals with how trajectories are used in this analysis.

According to the description in Section 2.3, a cluster of back trajectories is computed each day from an area within 30 km of the ship's position, on a 5 km grid.  The maps in Figure 2 show paths for back-trajectory clusters on two dates, illustrating what the authors describe as 'over ice' and 'open ocean - terrestrial' characteristics.  My main question is how the authors account for altitude variability in these computed trajectories, as this is not shown in a Figure or discussed in Section 2.3.

I believe these are isentropic back trajectories, and pressure altitude along the time steps should be one of the outputs.  The airmass will only be in contact with surface sources of CH4 if it is at the surface, within the boundary layer, which can be very shallow over the ice in the Arctic summer and is capped by a typically strong inversion.  Over land or open ocean the BL can be much deeper.   An airmass back trajectory in the free troposphere will not be influenced by surface emissions.

So, how exactly do the authors classify the significance of surface types and emissions over the back trajectory?  Do they exclude portions of the back trajectory that are above the BL?  If so, are you using estimates of BL depth from ERA5?  From the text in Section 2.3 it sounds like the surface type under the entire length of the 5-day trajectory is used to derive the classification and to identify potential sources of observed variability in the CH4 concentration.  But this cannot be a correct approach when significant portions of the trajectory are above the BL, and thus out of contact with the surface.  We need more explanation of the methodology here to understand the analysis.

**Response**

**Thank you very much for this important comment. We have now included the atmospheric boundary layer (ABL) height from the ERA5 reanalysis and considered only those trajectories with trajectory heights below the ABL. This analysis shows that 59% of the trajectory time steps during Leg 4 and 56% during Leg 5 were within the ABL. The remaining time steps, which were above the ABL, were excluded from the subsequent surface influence analysis (see revised Figs. 5–8). However, the overall correlations remained unchanged (Fig. 7 and 8). The modification is given in the Methods section (lines 133– 137).**

One final comment concerns the comparison with Arctic time series stations operated by NOAA/GML, AGAGE, etc.  It was helpful and informative to show some of this data on Figure 3.  The authors should be aware that NOAA/GML also took weekly flask samples at the Polarstern site during MOSAiC and these have been analyzed in their Boulder lab following the same rigorous protocols as other long-term GML monitoring sites.  It would be interesting to also see this data presented with your time series.

**Response**

**We have now included a comparison with the weekly flask data collected by NOAA/GML along with the year-long concentration dataset, in the Methods section (lines 88–89).**

---

## Author Response (AR2)

Dear Editor and Reviewer,

thank you for the final minor comment. We have addressed it in the revised manuscript (see track changes, lines 88-89).

We would like to thank the Reviewer once again for their valuable feedback and constructive suggestions throughout the review process. We hope that the manuscript is now ready for publication.

Best regards,
Amanda Sellmaier

---

## Author Response (AR3)

Dear Editor Team,

Thank you for accepting the manuscript for publication!
I am writing in response to the message **"Uploaded files validated" dated 06 November**.
**ROR entry:** The affiliation stated in the manuscript is correct. The corresponding authors are based at the *Alfred-Wegener-Institut Helmholtz-Zentrum für Polar- und Meeresforschung (Potsdam, Germany)*. However, since the ROR database does not list the Potsdam site, I selected the main institutional entry located in *Bremerhaven, Germany*.

I have now grouped the Appendix Figs within the Appendix section.

Best regards,
Amanda Sellmaier